# Comprehensive Genomic Analysis of *SnRK* in Rosaceae and Expression Analysis of *RoSnRK2* in Response to Abiotic Stress in *Rubus occidentalis*

**DOI:** 10.3390/plants12091784

**Published:** 2023-04-26

**Authors:** Guoming Wang, Sophia Lee Guan, Nan Zhu, Qionghou Li, Xinran Chong, Tao Wang, Jiping Xuan

**Affiliations:** 1Jiangsu Key Laboratory for the Research and Utilization of Plant Resources, Institute of Botany, Jiangsu Province and Chinese Academy of Sciences, Nanjing 210014, China; 2College of Computer, Mathematical, and Natural Sciences, University of Maryland, College Park, MD 20742, USA; 3Centre of Pear Engineering Technology Research, State Key Laboratory of Crop Genetics and Germplasm Enhancement, Nanjing Agricultural University, Nanjing 210095, China

**Keywords:** ABA, SnRK, abiotic stress, black raspberry, Rosaceae

## Abstract

The sucrose nonfermenting 1-related protein kinase (SnRK) plays an important role in responding to abiotic stresses by phosphorylating the target protein to regulate various signaling pathways. However, little is known about the characteristics, evolutionary history, and expression patterns of the *SnRK* family in black raspberry (*Rubus occidentalis* L.) or other Rosaceae family species. In this study, a total of 209 *SnRK* genes were identified in 7 Rosaceae species and divided into 3 subfamilies (*SnRK1*, *SnRK2,* and *SnRK3*) based on phylogenetic analysis and specific motifs. Whole-genome duplication (WGD) and dispersed duplication (DSD) were considered to be major contributions to the *SnRK* family expansion. Purifying selection was the primary driving force in the *SnRK* family evolution. The spatial expression indicated that the *RoSnRK* genes may play important roles in different tissues. In addition, the expression models of 5 *RoSnRK2* genes in response to abiotic stresses were detected by qRT-PCR. The proteins encoded by *RoSnRK2* genes localize to the cytoplasm and nucleus in order to perform their respective functions. Taken together, this study provided an analysis of the *SnRK* gene family expansion and evolution, and contributed to the current knowledge of the function of 5 *RoSnRK2* genes, which in turn expanded understanding of the molecular mechanisms of black raspberry responses to abiotic stress.

## 1. Introduction

Plants develop various molecular defense mechanisms to adapt to biotic and abiotic stresses, including heat, cold, waterlogging, drought, and salinity stresses. To cope with these stresses, plants establish a network of important defense-related metabolic mechanisms through the regulation of gene expression and protein modification [1]. Protein kinase-mediated phosphorylation and dephosphorylation play an important role in protein modification to respond to stress signals [2]. The various reported protein kinase genes that are related to resistance include mitogen-activated protein kinase (MAPK) [3], calcium dependent protein kinase (CDPK) [4], receptor-like kinase (RLK) [5,6], and sucrose nonfermenting 1 (SNF1)-related protein kinase (SnRK) [7]. In addition to signaling networks and developmental processes, plant protein kinases regulate stress response. As a result of understanding plant responses to diverse stresses, genetic engineering can be used to improve plant tolerability to abiotic stresses, thus enabling the advent of new agricultural practices [5,6].

The SnRK family is a serine/threonine (Ser/Thr) family of protein kinases, which specifically and widely exists in plants. In plants, SnRKs protein kinases play vital roles in regulating metabolism and stress responses. The SnRK family diverges into 3 subfamilies based on their phylogenetic relationships and gene structures, including SnRK1, SnRK2 and SnRK3 [8]. The SnRK1 subfamily has a highly conserved N-terminal Pkinase domain, and they are most similar to SNF1 and AMPK in structure and function [9]. The subfamilies SnRK2 and SnRK3 are unique to plants, and both are diverse and have more members than the SnRK1 subfamily. The SnRK2 subfamily contains a conserved Pkinase domain and a C-terminal variable adjusting conserved domain, while the SnRK3 subfamily harbors N-terminal Pkinase domains and NAF domains in the C-terminal [10].

The SnRK1 subfamily is also involved in signaling pathways that include metabolic stress, cell cycle control, pathogen responses, and meristem formation [8,11]. The SnRK1 subfamily possesses relatively few members, although these were identified in many plant species. In *Arabidopsis thaliana*, SnRK1 is involved in the *IDD8*-mediated sugar metabolic pathway of flowering time under low-sugar conditions [12]. In barley, the SnRK1 protein kinase was shown to be involved in regulating pollen development and can lead to infertility [13]. In *Solanum tuberosum*, SnRK1 is involved in carbohydrate metabolism regulation [14].

Unlike the SnRK1 subfamily, SnRK2 and SnRK3 were evolved from the SnRK1 subfamily via gene duplication in plant evolution [8]. The SnRK2 subfamily has Domain II with about 40 amino acid residues that can mediate the interaction with the clade A type 2C protein phosphatases (PP2Cs) and play a significant role in abscisic acid (ABA)-mediated responses to abiotic stresses, especially for osmotic and salt stresses [15]. In the ABA signaling pathway, the ABA receptors Pyrabactin Resistance 1 (PYR1)/PYR1-like (PYL) interact with the clade A type of PP2Cs. In the absence of ABA, PYL protein cannot bind to PP2Cs to prevent SnRK2 activation, whereas ABA-bound PYR/PYL can interact with PP2Cs and inhibit PP2Cs from dephosphorylating SnRK2 [16,17]. In turn, phosphorylated SnRK2 activates downstream targets, which include ABA-responsive element-binding factors (AREBs/ABFs) as well as other targets [18]. AtSnRK2.4 and AtSnRK2.10 modulates the root system architecture in response to salinity by influencing the transcript levels of aquaporins in *Arabidopsis thaliana* [19]. AtSnRK2.10 phosphorylates dehydrins ERD10 and ERD14 to deal with the osmotic stress in *Arabidopsis thaliana* [20]. The overexpression of *NtSnRK2.1* enhances salt tolerance in transgenic *Nicotiana tabacum* [21]. AtSnRK2.6 plays an important role in stomatal guard cells in ABA signaling [22]. In other species such as maize [23], rice [24], wheat [25], and cotton [26], most *SnRK2* genes respond to more than one type of stress, and the overexpression of *SnRK2* genes could increase the tolerance to abiotic stress. These findings suggest that genes of the SnRK2 subfamily are activated by abiotic stress and play major roles in multiple abiotic stress responses.

The SnRK3 kinases, designated as calcineurin B-like calcium sensor-interacting protein kinases (CIPKs), interact with calcium sensor calcineurin B-like proteins (CBLs) to mediate the calcium signaling pathway and various stresses in plants [10,27]. The CBL-CIPK protein complex enables information integration and physiological coordination to resist various stresses in plants [28]. CIPKs play important roles in the response to salt tolerance in plants [29,30,31]. The growing body of the data reveals the high relevance of *SnRK* function in nutritional efficiency and stress responses, and ultimately in improving plant tolerance to multiple abiotic stresses.

Although the *SnRK* gene family has a crucial role in the abiotic stress response and the ABA signaling pathway, the *SnRK* family has not yet been systematically investigated in Rosaceae, which is an economically important family that includes many best-selling commercial species and has available genomes, such as pear, apple, strawberry, peach, sweet cherry, Japanese apricot, and black raspberry. Black raspberry (*Rubus occidentalis*) is an important fruit native to eastern North America and possesses a unique flavor, is a rich source of antioxidants, and exhibits potential health benefits. Studies on functional *SnRK* genes in black raspberry are still limited. In this study, a total of 209 *SnRK* genes were identified in 7 Rosaceae species, allowing for the inference of their phylogenic relationships, expansion, and evolution. The expression patterns of 5 *RoSnRK2* genes in black raspberry were explored under salt and ABA treatments. Overall, this study provides a basis for elucidating the roles of *RoSnRK2* genes in the control of plant growth and the response to abiotic stress.

## 2. Results

### 2.1. Identification, Phylogenetic Tree and Classification of SnRK Genes in 7 Rosaceae Species

To investigate the *SnRK* gene family in 7 Rosaceae species, two methods were applied, including BLASTp and HMM. A total of 38 *AtSnRK* genes in *Arabidopsis* [10] were used as queries to BLASTp in 7 Rosaceae species’ genomes with E-value < 1 × e^−10^. After further screening of the *SnRK* sequences, a total of 209 *SnRK* genes were identified in 7 Rosaceae species, including 23 in black raspberry, 38 in Chinese white pear, 46 in apple, 26 in peach, 25 in Japanese apricot, 26 in strawberry, and 26 in sweet cherry. The sequence feature of the SnRK proteins is shown in Appendix A, including gene position in chromosomes, lengths of gene sequences, protein molecular weight, and isoelectric point.

The phylogenetic tree of the SnRK proteins of 7 Rosaceae species was constructed with the maximum likelihood method (ML) algorithm to classify the *SnRK* genes and investigate the evolutionary relationships. All the SnRKs were specifically classified into 3 separated clades of SnRK1, SnRK2 and SnRK3 subgroups (Figure 1), which were have been reported earlier in *Arabidopsis* [10]. Of these SnRK proteins, 209 have complete functional domains, including 9 proteins in the SnRK1 subfamily with Pkinase (PF00069), UBA (PF00627) and KA1 (PF02149) domains; 142 proteins in the SnRK3 subfamily with Pkinase and NAF (PF03822) domain; and 58 proteins in the SnRK2 subfamily with Pkinase domains in addition to a strong homology to the AtSnRK2 subfamily. The SnRK3 subfamily includes a large number of SnRK3, while the SnRK1 subfamily has a small number of SnRK1. The genes of *SnRK* in the same clades may perform similar functions. The *SnRK2* subfamily includes essential components and specific positive regulators of the ABA signaling pathway.

### 2.2. Chromosome Location and Synteny Analysis of SnRK Gene Family

To identify the origin of *SnRK* family members, 5 modes of gene duplication were identified and analyzed in 7 Rosaceae species, including whole-genome duplication (WGD) or segmental duplication, tandem duplication (TD), proximal duplication (PD), transposed duplication (TRD), and dispersed duplication (DSD). A total of 289 duplicated gene pairs were found in the *SnRK* family members and classified into 5 duplication modes (Figure 2 and Appendix A). In addition, a large number of the *SnRK* gene pairs in all investigated species were assigned to WGD (27.0%) and DSD (62.3%), and 10.7% duplicated gene pairs assigned to TD, PD and DSD. WGD and DSD impacted the evolution of the *SnRK* family. Genomic rearrangements or gene loss may have led to the large proportion of DSD in Rosaceae species. The number of WGD in apple, Chinese white pear, black raspberry, Japanese apricot, strawberry, peach, and sweet cherry were 30, 14, 8, 6, 6, 6, and 3, respectively. Compared to the number of WGD, the number of DSD in turn were respectively 36, 32, 19, 23, 22, 21, and 19, respectively (Figure 2 and Appendix A). The results show that DSD was ubiquitous in 7 Rosaceae species.

A total of 209 *SnRK* genes were anchored onto the chromosomes of each species based on the genome annotations. The *SnRK* genes were found to be randomly distributed on each chromosome. In addition, we investigated the 79 collinearity gene pairs of *SnRK* in each Rosaceae species, including 32, 15, 8, 6, 6, 6, and 6 pairs in apple, Chinese white pear, black raspberry, strawberry, Japanese apricot, peach, and sweet cherry, respectively (Figure 3, Appendix A). Furthermore, the collinearity of the *SnRK* gene pairs between black raspberry and the other six Rosaceae species were investigated. A total of 194 gene pairs of collinearity were identified including between black raspberry and strawberry (26), apple (45), sweet cherry (31), Japanese apricot (30), peach (27), and Chinese white pear (35) (Appendix A). The results revealed a strong collinearity relationship between black raspberry and the other six investigated species.

### 2.3. SnRK Genes Evolved under Strong Purifying Selection

To further elucidate the evolutionary trajectory of the *SnRK* genes in Rosaceae, we performed a comparative analysis of the dynamic evolution for different model gene duplications. Here, the Ka, Ks and Ka/Ks ratios for the duplicated pairs were estimated to measure the divergence and direction of selectional pressure in coding sequences (Figure 4 and Appendix A). Ka/Ks > 1 indicated positive selection (Darwinian selection); Ka/Ks = 1 indicated neutral evolution; and Ka/Ks < 1 indicated negative (purifying) selection [32]. The deleterious mutations were removed by negative selection; on the contrary, new advantageous mutations were accumulated by positive selection [33]. All Ka/Ks ratios of the *SnRK* paralogous genes in the 7 Rosaceae species were less than one, suggesting that purifying selection was the primary driving force on the *SnRK* family genes.

### 2.4. The Analysis of Gene Structure and Protein Characterization

All RoSnRK protein sequences in black raspberry were used to construct a phylogenetic tree. The 23 RoSnRK were named as RoSnRK1.1-1.2, RoSnRK2.1-2.5, and RoSnRK3.1-3.16 according to phylogenetic analysis and multiple sequence alignment with *Arabidopsis* (Figure 1 and Figure 5). We explored GSDS to determine the intron/exon structure of *RoSnRK* genes (Figure 5B). The *RoSnRK1* subfamily genes have 11 introns, while the *RoSnRK2* subfamily contains 5 to 8 introns. However, the number of introns in the *RoSnRK3* subfamily varies. The genes of *RoSnRK3.1*-*3.10* contained less than 3 introns, and the genes of *RoSnRK3.11*-*3.16* contained 9 to 14 introns. The combined phylogenetic tree and intron/exon structure analysis of the *RoSnRKs* showed that members of the same subfamily exhibited close evolutionary relationships and similar organizations. The cis-elements in promoter regions play a critical role in the plant signal transduction by being bound by their cognate transcription factors. Therefore, the promoter sequences 2.0 kb upstream of the *RoSnRK* genes were analyzed by using PlantCARE. Some common cis-regulatory elements related to light, transcriptional regulation factor, plant hormone, and regulatory stress responses were briefly marked in the promoter regions, and the diversity of the cis-elements in the *RoSnRK* genes showed that the expression may differ in response to development and abiotic stress (Appendix A).

RoSnRK protein sequences in black raspberry were further analyzed using MEME by choosing the 15 motifs with default parameters (Figure 5C). The results showed that the same subfamilies have similar motifs, while different subfamilies retained an obvious difference in motif composition. Motifs 6, 7, and 8 were present in all RoSnRKs, while motifs 12 and 15 only appeared in the RoSnRK3 subfamily. The SnRK1 and SnRK2 subfamilies share a common type and position of the conserved motif. In addition, the conserved domain of RoSnRK was analyzed by SMART. All the proteins of RoSnRK have a highly conserved N-terminal Pkinase domain (Figure 5D). The proteins of the RoSnRK1 subfamily have a conserved KA1 domain and the proteins of the RoSnRK3 subfamily have conserved NAF domain in the C-terminal (Figure 5D).

### 2.5. Expression Analysis of RoSnRK Genes in Different Tissues of Black Raspberry

To study the expression levels and functional properties of the *SnRK* genes in different tissues, we investigated the spatial expression of the *RoSnRK* genes in roots, canes, leaves, and fruits at different developmental stages in black raspberry. The expression patterns of 23 *RoSnRK* genes were analyzed with published RNA-seq data [34]. These *RoSnRK* genes showed different expression patterns in different tissues, and they may have different functions (Figure 6 and Appendix A). The expression levels of *RoSnRK* family genes in roots were generally highly expressed, except for the genes *RoSnRK3.3*, *RoSnRK3.4*, *RoSnRK3.11*, and *RoSnRK3.16.* Most of the *RoSnRK* genes presented low expression levels in mature leaves, whereas the genes *RoSnRK3.3*, *RoSnRK3.4*, *RoSnRK3.7*, and *RoSnRK3.9* exhibited higher expression levels in mature leaves compared to in other tissues. The expression abundance of the genes *RoSnRK2.1*, *RoSnRK3.15*, and *RoSnRK3.16* increased with increasing fruit maturity. Conversely, with increased fruit maturity, the gene expression levels of *RoSnRK3.3*, *RoSnRK3.8*, and *RoSnRK3.11* decreased. The results revealed that the *RoSnRK* genes may play important roles in various black raspberry tissues and display different expression characteristics in different fruit development stages.

### 2.6. Expression Analysis of RoSnRK2 Genes under Different Abiotic Stresses

Many studies have shown that the members of the *SnRK2* subfamily play key roles in the response to multiple abiotic stresses such as osmotic stress, salinity, drought, and exogenous ABA [15]. The expression patterns of 5 *RoSnRK2* subfamily genes in black raspberry were detected by qRT-PCR, and the results showed that the *RoSnRK2* genes exhibited transcriptional changes under NaCl or ABA treatment, suggesting that the response of the *RoSnRK2* genes to multiple stresses is a dynamic process (Figure 7). All 5 *RoSnRK2* genes were activated under the NaCl treatment at each time point except for *RoSnRK2.4* at 24 h (Figure 7A). *RoSnRK2.1* and *RoSnRK2.4* peaked at 6 h, and *RoSnRK2.5* peaked 12 h after the NaCl treatment. The expression of *RoSnRK2.4* first increased, then decreased. After exogenous ABA treatment, *RoSnRK2.1* and *RoSnRK2.2* showed no significant difference at 6 h, then gradually increased significantly over time and peaked at 24 h (Figure 7B). The expression patterns of 3 genes (*RoSnRK2.3*, *RoSnRK2.4*, and *RoSnRK2.5*) showed significant differences at 6 h and showed a wavy trend under the ABA treatment. Taken together, all 5 *RoSnRK2* genes were induced under the NaCl or ABA treatments. The diverse expression pattern suggests that the *RoSnRK2* genes may be critical to abiotic stresses.

### 2.7. Subcellular Localization of RoSnRK2 Proteins

Previous research has shown that the SnRK2 proteins were mainly localized to the cytoplasm and nucleus [35,36]. To determine the subcellular localizations of the RoSnRK2 proteins, 35S-RoSnRK2s-GFP constructs were introduced into tobacco (*Nicotiana benthamiana*) cells. The fluorescences of GFP fused with RoSnRK2.1-5 were detected in both the cytoplasm and the nucleus (Figure 8). The proteins encoded by the genes localized to specific cell organelles to perform their respective functions. In the ABA signaling pathway, the PP2Cs interact with and inactivate the ABA-activated SnRK2 via dephosphorylation, although the PP2C proteins are localized exclusively in the nucleus [37,38].

### 2.8. Protein–Protein and Protein–Chemical Interaction Analysis

To explore the potential molecular mechanisms of RoSnRKs, the protein–protein interaction network was constructed in the STRING database. The results predicted that the interaction of RoSnRK proteins with 5 different proteins: SNF4 (Homolog of yeast sucrose nonfermenting 4), OZS1 (C4-dicarboxylate transporter/malic acid transport protein), HAB1 (Protein phosphatase 2C 16), CBL1 (Serine/threonine-protein phosphatase 2b regulatory subunit), and ABI1 (Protein phosphatase 2C family protein) (Figure 9A). In addition, the protein–chemical interaction network was performed in the STITCH database. The analysis suggested the interaction of the RoSnRK proteins with four different chemicals, including MgATP (507.2 g/mol), MgADP (427.2 g/mol), adenosine monophosphate (347.2 g/mol), and calcium ions (Figure 9B). It is noteworthy that calcium plays a vital role in the anatomy, physiology and biochemistry of organisms and of the cell, particularly in signal transduction pathways. In short, the RoSnRK proteins play important roles in the organism.

## 3. Discussion

The SnRKs regulate various metabolic pathways in plants, including gene expression, protein synthesis, and carbohydrate metabolism. They are vital in plants and play a crucial role in various metabolic pathways under abiotic stress. They are a group of protein kinases that are activated under several abiotic stresses, such as drought, salinity, and extreme temperatures. The primary function of *SnRK* family genes is to regulate the energy balance of plant cells, allowing them to adapt and survive in adverse environmental conditions [8,39]. The identification and function of the *SnRKs* genes have been studied in many species. However, the the *SnRK* gene family has not been studied widely in the Rosaceae family, although *SnRK2* subfamily was identified and analyzed in pear [40]. In this study, the genome-wide identification and comparative analysis of the *SnRKs* genes from 7 Rosaceae species were carried out, and the potential role of *SnRKs* in black raspberry was investigated. A total of 209 *SnRK* genes were identified in 7 Rosaceae species and subsequently 209 *SnRK* genes diverged into 3 clusters: *SnRK1*, *SnRK2*, and *SnRK3,* based on the phylogenetic analysis with *Arabidopsis* homologs and conserved motifs structures, which is consistent with the classification in other plant species [10]. All *SnRK* family members retain an N-terminal protein kinase domain, but various *SnRK* subfamilies contain different conserved domains at the C-terminal [8,24]. For example, the *SnRK1* subfamily contains UBA and KA1 domains in the C-terminal, the *SnRK2* subfamily harbors variable adjusting conserved domains, and the *SnRK3* subfamily contains a NAF domain [10]. The 209 *SnRK* genes exhibited a widespread and uneven distribution across chromosomes. In black raspberry, twenty-three *SnRK* genes were categorized into two *RoSnRK1* genes, 5 *RoSnRK2* genes, and sixteen *RoSnRK2* genes. Different subfamily genes showed discrepant gene lengths and exon intron structural divergences. Structural features showed that the same subfamily exhibited close phylogenetic relationships and similar organizations. The genes with less introns were identified to have higher expression in plants [41]. All twenty-three RoSnRK proteins contained motif 6, 7, and 8, indicating a highly conserved function. The genes in the same subfamilies have similar motifs, while different subfamilies retained obvious differences in motif composition. *SnRK1* plays a crucial role in the regulation of carbohydrate metabolism, cell cycle, and meristem development [8,14], while *SnRK2* and *SnRK3* were shown to perform vital functions in the resistance to various stresses [10]. The *RoSnRK* genes of the same subfamily have similar specialized biological functions, but their specific functions require further investigation.

Generally, gene duplication events drive gene family expansion in eukaryotes and include 5 types (WGD, TD, PD, DSD and TRD) [42]. Gene duplication can lead to new gene functions and evolutionary processes [43]. WGD events can produce a large number of duplicate genes in a relatively short time [44]. WGD is the main driving force of new gene functions in the genetic evolutionary system [42]. For example, the *F-box* and *PP2C* gene families in pear expanded primarily through WGD or DSD duplication events [38,45]. In our study, we found that WGD and DSD were the main driving forces for the expansion of the *SnRK* gene family in 7 Rosaceae species (Figure 2). In addition, we found that most Ka/Ks ratios of *SnRK* gene pairs in 7 Rosaceae species were less than one, suggesting that the genes of the *SnRK* family have experienced strong purifying selection.

Analyzing *RoSnRK* gene expression patterns across various tissues, developmental stages, and stress responses will further the current understanding of gene functions. All the *RoSnRK* genes showed different expression patterns in different tissues and developmental stages, indicating that these genes may play important roles in black raspberry. For example, 5 *RoSnRK2* subfamily genes were uniformly highly expressed in roots, suggesting that root tissues may respond quickly to plant stress. In *Arabidopsis* roots, *SnRK2.4* and *SnRK2.10* were verified as positive regulators of root growth under salt stress [46]. The *SnRK2.4* gene was expressed in most tissues in the main root, whereas the expression of *SnRK2.10* was identified in the vasculature at the sites of lateral root formation [46]. Under abiotic stress, SnRKs act as a primary sensor and signal transducer in plants. They activate signaling pathways that contribute to the adaptation of plants to stress conditions. SnRK-induced changes in gene expression lead to the production of stress-responsive proteins, including osmoprotectants, heat shock proteins, and antioxidative enzymes, which protect the plant from damage caused by environmental stress [8]. For *SnRK2* subfamily genes, extensive studies show that they contribute significantly to plant responses to stress [20,21,23,24]. For example, *SnRK2s* are activated under drought stress and regulate stomatal closure, leading to a reduction in water loss via transpiration [47]. Moreover, *SnRK2* genes play crucial roles in the regulation of ABA signal transduction [48]. In this study, 5 *RoSnRK2* subfamily genes were screened to detect expression levels by qRT-PCR under the NaCl or ABA treatment. Most *RoSnRK2* genes exhibited high expression during the salt and ABA treatments, showing that *RoSnRK2* expression is sensitive to salt and ABA in black raspberry (Figure 7). However, different *RoSnRK2* subfamily genes had different responses to these treatments, which was consistent with the highly complex regulatory network of plants under stress treatments. In conclusion, *SnRKs* play a vital role in plants under abiotic stress conditions to maintain energy balance and regulate different metabolic pathways. Ultimately, the activation and regulation of the *RoSnRK2* subfamily contribute to the survival and adaptation of plants under adverse environmental conditions.

## 4. Materials and Methods

### 4.1. Identification of SnRK Genes in 7 Rosaceae Species

The genome sequence of black raspberry (*Rubus occidentalis*), apple (*Malus domestica*), strawberry (*Fragaria vesca*), peach (*Prunus persica*), and sweet cherry (*Prunus avium*) genomic datasets were retrieved from the Rosaceae Genome Database (http://www.rosaceae.org, accessed on 1 June 2020). The genome sequence of the Chinese white pear (*Pyrus bretschneideri*) was downloaded from the Pear Genome Project (http://peargenome.njau.edu.cn/, accessed on 1 June 2020). The genome sequence of the Japanese apricot (*Prunus mume*) was collected from the *Prunus mume* Genome Project (http://prunusmumegenome.bjfu.edu.cn/index.jsp, accessed on 1 June 2020). The locus IDs of *SnRK* genes of *Arabidopsis thaliana* were listed in Appendix A and the sequences of *SnRK* genes were downloaded from TAIR (http://www.arabidopsis.org/, accessed on 1 June 2020) [10]. The Hidden Markov Model (HMM) and the BLASTP program were performed for the preliminary identification of SnRK proteins. The Pfam protein family database (http://pfam.sanger.ac.uk/, accessed on 1 June 2020) was used to download the HMM files of each subfamily of SnRKs, and HMMER 3.0 was used to search the dataset for SnRKs based on the default threshold. The *SnRK* genes of *Arabidopsis thaliana* were used as queries in a TBLASTN (version 2.2.26, Bethesda, Rockville, MD, USA) search against the 7 Rosaceae species genomes with an E-value threshold of 1 × e^−10^. All candidate sequences of *SnRK* genes were further identified based on the Pfam database (http://pfam.xfam.org/, accessed on 1 April 2022) and the SMART database (http://smart.embl-heidelberg.de/, accessed on 1 April 2022). Moreover, Protein Isoelectric Point (pI) and Protein Molecular Weight (MW) of each SnRK protein were calculated by online tools (http://www.bioinformatics.org/sms2/protein_iep.html, accessed on 1 January 2023).

### 4.2. Phylogenetic Tree and Gene Structure Analysis

Full length protein sequences of SnRK alignment were performed using MUSCLE with default parameters in MEGA7.0 [49]. The maximum likelihood method (ML) with the Jones–Taylor–Thornton (JTT) model was constructed with bootstrap 1000 to assess the statistical support for each node. The exon–intron organization of the *RoSnRK* genes was analyzed by the Gene Structure Display Server (GSDS: http://gsds.cbi.pku.edu.cn/, accessed on 1 January 2023). The conserved motifs of RoSnRKs were analyzed by Motif Elicitation (MEME: http://meme.sdsc.edu/meme/itro.html, accessed on 1 January 2023). The conserved domain of RoSnRKs was analyzed by SMART.

### 4.3. Cis-Element Predictions of RoSnRKs

All *RoSnRKs* promoter sequences (2000 bp upstream from initiation codons) were isolated from genome of *Rubus occidentalis*. The cis-regulatory elements of *RoSnRKs* were predicted by the PlantCARE database (http://bioinformatics.psb.ugent.be/webtools/plantcare/html, accessed on 25 January 2023). The results were visualized with TB-tools software v1.1 [50].

### 4.4. Location of SnRK Genes on Chromosomes and Synteny Analysis

The chromosomal location information of *SnRK* genes was determined from genome annotation data of 7 Rosaceae genome projects. The location data of *SnRK* genes were plotted using TB-tools software v1.1 [50]. The method similar to that used for the PGDD (http://chibba.agtec.uga.edu/duplication/, accessed on 25 January 2023) was performed to analyze the synteny among 7 Rosaceae genomes [51]. Then, BLASTP was performed for potential homologous gene pairs of SnRK protein sequences (E < 1 × 10^–5^, top 5 matches). The orthologous gene pairs of *SnRK* were determined by MCScanX within and between each Rosaceae species, and then the duplicated *SnRK* genes were classified into the following models: whole genome duplication (WGD)/segmental, tandem, transposed, proximal, and dispersed [52]. The results were displayed by TB-tools software v1.1 [50].

### 4.5. Calculating Ka, Ks, and Ka/Ks of Duplicated Gene Pairs

MCScanX downstream analysis tools were performed to annotate the Ka (non-synonymous) and Ks (synonymous) substitution rates of syntenic gene pairs. The Ka, Ks, Ka/Ks, and *p*-value were generated using KaKs Calculator 2.0 with Nei–Gojobori (NG) method [53].

### 4.6. Expression Analysis of RoSnRKs in Different Tissues

The raw sequence data of different tissues were obtained from the NCBI (National Center for Biotechnology Information) database (https://www.ncbi.nlm.nih.gov/, accessed on 25 January 2023) under the accession number PRJNA430858. By removing low-quality reads, poly (A/T) tails, and adapter sequences from raw reads, the raw reads were cleaned. Clean reads were aligned to the reference genome using HISAT2 and feature counts were performed to estimate transcript abundance levels. Finally, the expression levels of *RoSnRK* genes were determined by fragments per kilobase million (FPKM). The RPKM values were calculated and used to estimate the gene expression abundances. The results were visualized though heatmap by the TB-tools software v1.1.

### 4.7. Plant Materials Treatment and qRT-PCR Analysis

The black raspberry ‘Bristol’ was planted in a growth chamber (photoperiod 16/8 h, the temperature 25 ± 1 °C, and 75% relative humidity) at the Institute of Botany, Jiangsu Province and Chinese Academy of Sciences, Nanjing. The raspberry seedlings were exposed to salt and ABA stresses, respectively. The plants were cultured in 200 mM NaCl for salt treatment, and 100 μM ABA for ABA treatment. All treated samples were collected in time intervals of 0, 6, 12, and 24 h, respectively. The black raspberry leaves were immediately frozen in liquid nitrogen and stored at −80 °C for RNA extraction. Total RNA was extracted using the RNAprep Pure Plant Kit (Tiangen, Beijing, China). The total RNAs were reverse-transcribed using TransScript One-Step gDNA Removal and cDNA synthesis Supermix (TransGen, Beijing, China). The primers of *RoSnRK* genes were designed using Primer Premier 6.0 and provided in Appendix A. *Ru18S* was used as the reference gene [54]. qRT-PCR was performed using a LightCycler 480 SYBRGREEN I Master (Roche, Diagostics, Rotkreuz, Switzerland). All reactions were carried out with 3 independent biological replicates. The relative expression levels were calculated with the 2^−ΔΔCt^ method. The results were calculated using Office 2010, and statistical analyses were performed by applying Student’s *t*-test at the levels of significant difference (*p* < 0.05 and *p* < 0.01, respectively).

### 4.8. Subcellular Localization

For the subcellular localization analysis, full-length CDS sequences without the termination codon of *RoSnRK2s* were fused into the vector pCAMBIA1300-35S: CDS-GFP. Primers used for cloning are listed in Appendix A. The recombinant plasmids and control plasmids were transformed into the GV3101 strain of *Agrobacterium tumefaciens*. The infection solution (10 mM MgCl_2_, 10 mM MES, and 100 mM acetosyringone, pH = 5.6) with suspended *Agrobacterium tumefaciens* (OD600 = 0.8) were injected into 30-day-old tobacco (*Nicotiana benthamiana*) leaves. Two days later, fluorescence was visualized using a confocal microscope LSM780 (Zeiss LSM 780, Oberkochen, Germany).

### 4.9. Protein–Protein and Protein–Chemical Interaction Analysis

The protein–protein interaction was predicted using the STRING (https://cn.string-db.org/, accessed on 25 March 2023) and protein–chemical interaction was predicted using STITCH (http://stitch.embl.de/, accessed on 25 March 2023) servers according to the methods described in the reference [55,56]. Furthermore, interaction networks were constructed by Cytoscape software (https://cytoscape.org/download.html, accessed on 25 March 2023).

## 5. Conclusions

In this study, a total of 209 *SnRK* genes were identified in 7 Rosaceae species. WGD and DSD were major contributions to *SnRK* family expansion. Purifying selection was the primary driving force in the *SnRK* family evolution. The spatial expression of the *RoSnRK* genes may play important roles in various black raspberry tissues. The diverse expression pattern suggests that the *RoSnRK2* genes may be critical to abiotic stress. The proteins encoded by *RoSnRK2* genes localize to the cytoplasm and nucleus in order to perform their respective functions. Consequently, our findings provide a foundation for *SnRK* gene family expansion and evolution, and the potential function of 5 *RoSnRK2* genes, which in turn expanded the understanding of the molecular mechanisms of black raspberry under various stress conditions.

## Figures and Tables

**Figure 1 plants-12-01784-f001:**
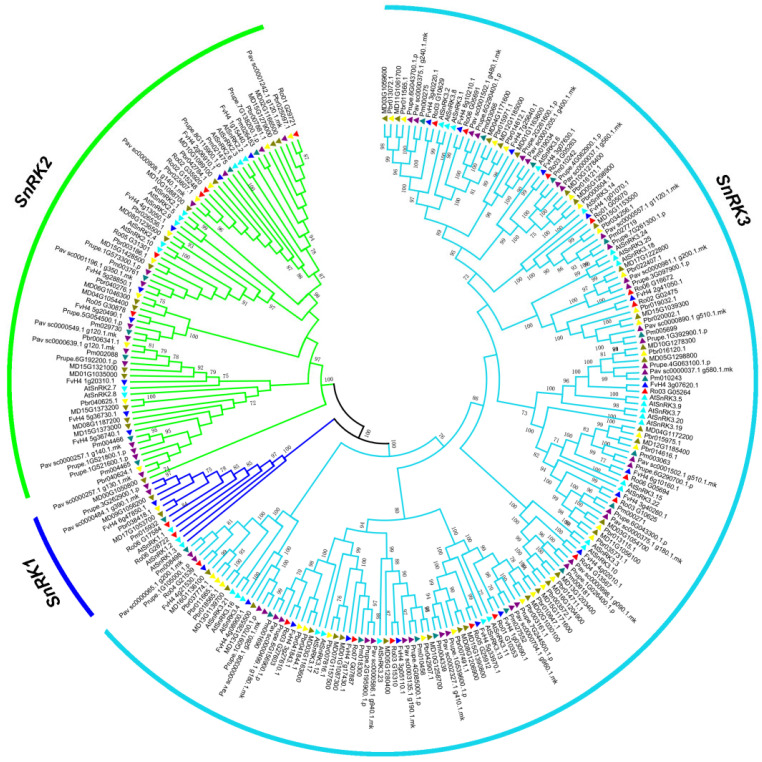
Phylogenetic relationship of SnRK of Rosaceae species and *Arabidopsis*. The phylogenetic tree was constructed with the maximum likelihood method. The blue, green, and light blue lines depict the SnRK1, 2, and 3 subfamilies, respectively.

**Figure 2 plants-12-01784-f002:**
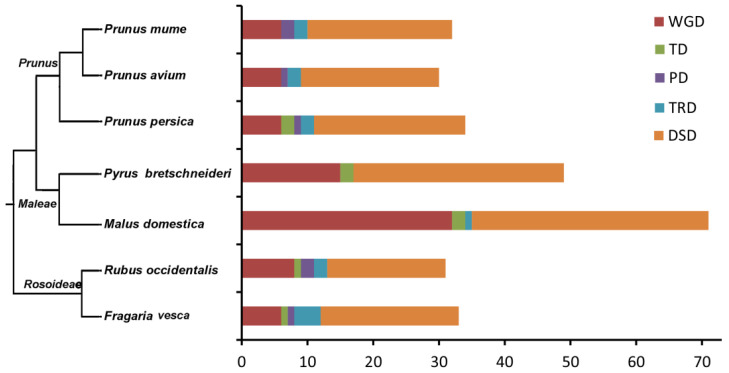
The number of *SnRK* gene pairs derived from different modes of gene duplication in 7 Rosaceae species. The phylogenetic relationship among 7 Rosaceae fruit species. The number of different modes of duplicated gene pairs in each species. WGD: whole genome duplication, TD: tandem duplication, PD: proximal duplication, TRD: transposed duplication, DSD: dispersed duplication.

**Figure 3 plants-12-01784-f003:**
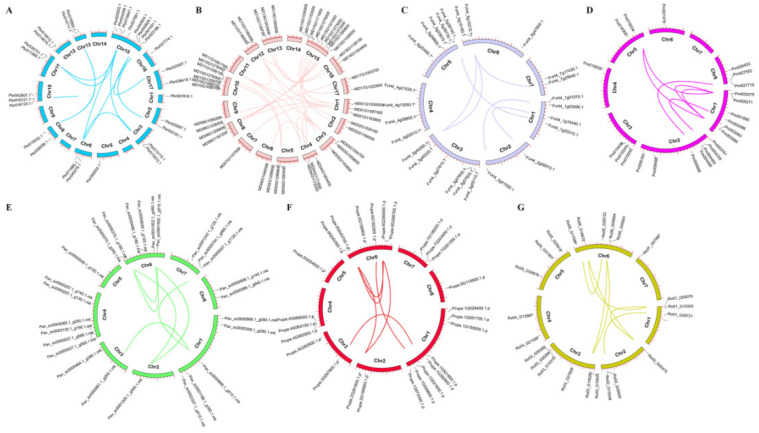
Chromosomal localization and syntenic relationships of *SnRK* genes in 7 Rosaceae species. (**A**) Chinese white pear; (**B**) apple; (**C**) strawberry; (**D**) Japanese apricot; (**E**) sweet cherry; (**F**) peach; (**G**) black raspberry. *SnRK* genes are mapped on different chromosomes and syntenic gene pairs are linked by colored lines.

**Figure 4 plants-12-01784-f004:**
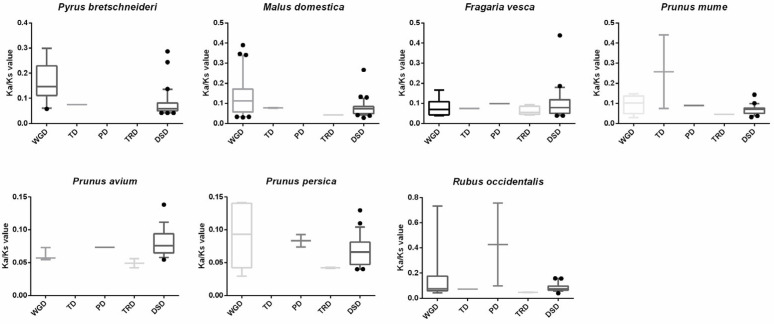
Ka/Ks distribution of different modes of duplicated *SnRK* gene pairs in 7 Rosaceae species. The x-axis represents 5 different duplication categories. The y-axis indicates the Ka/Ks ratio.

**Figure 5 plants-12-01784-f005:**
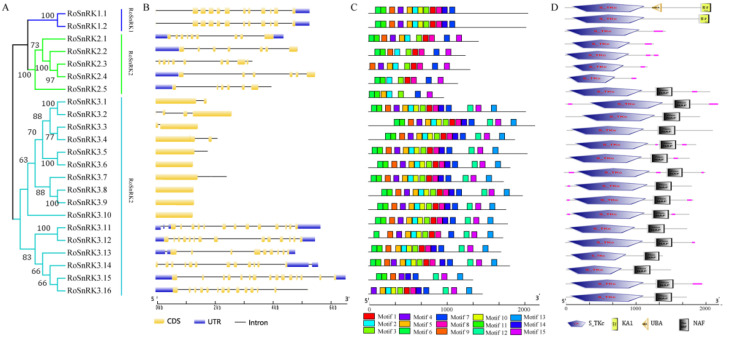
Phylogenetic relationships, gene structure and conserved protein motifs of the *SnRK* genes in *Rubus occidentalis*. (**A**) Phylogenetic tree of 23 RoSnRK proteins. (**B**) Exon/intron and UTR organization of *RoSnRK* genes. (**C**) The motif composition of RoSnRK proteins. Motifs are displayed in different colored boxes. (**D**) The conserved domain composition of RoSnRK proteins. Protein lengths can be estimated using the scale at the bottom.

**Figure 6 plants-12-01784-f006:**
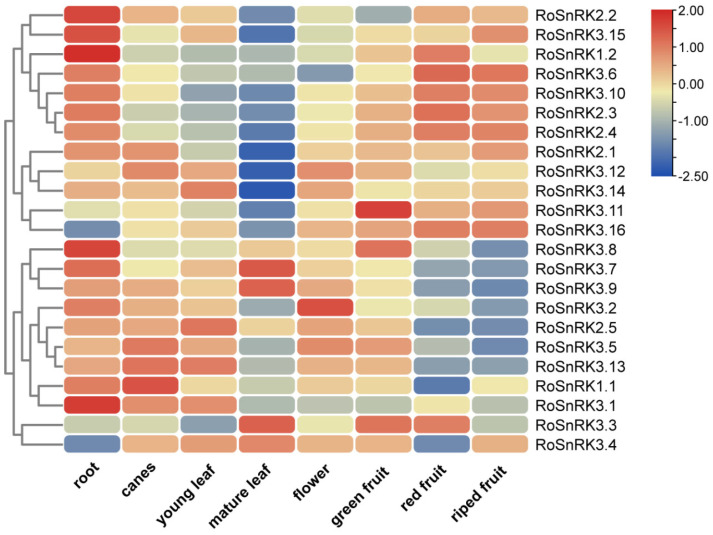
Expression patterns of 23 *RoSnRK* genes in different tissues. Expression values are plotted as log_2_ transformed FPKM. Expression levels are indicated by the color bar, with red indicating high expression and blue indicating low expression.

**Figure 7 plants-12-01784-f007:**
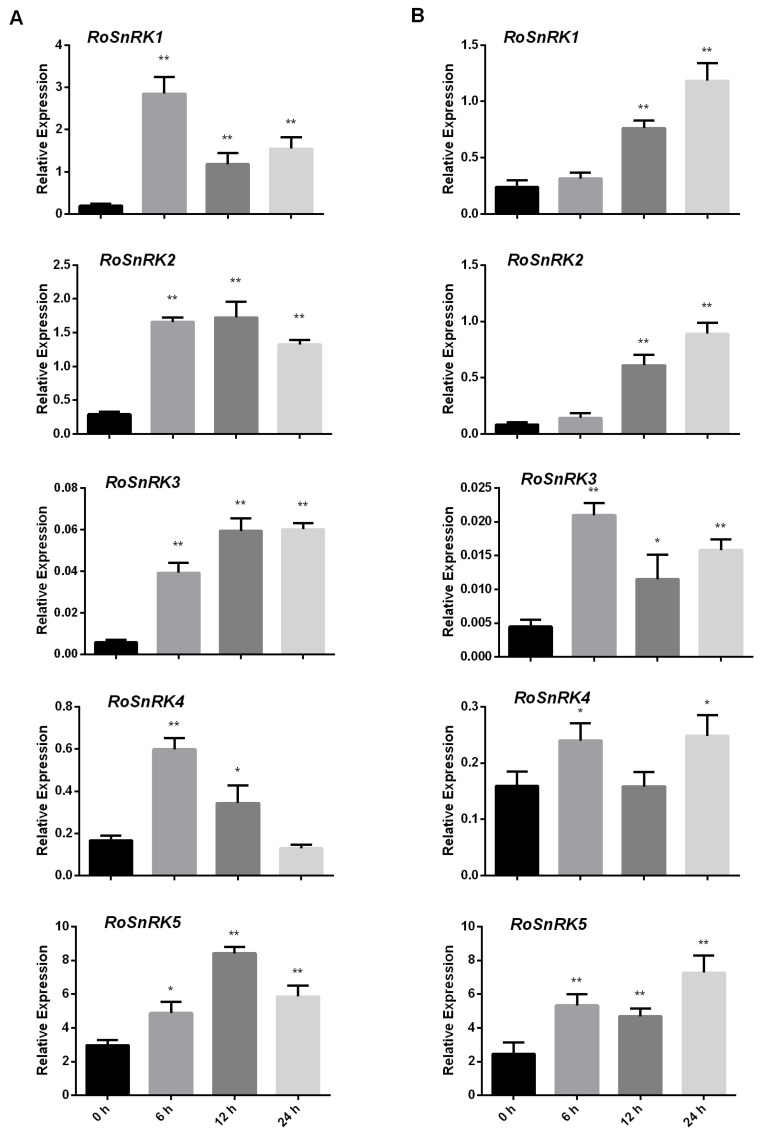
qRT-PCR analysis of *RoSnRK2* genes under (**A**) NaCl and (**B**) ABA treatments. Standard errors and ANOVA were calculated by applying Student’s *t*-test. Single and double stars stand for the levels of significant difference (*p* < 0.05 and *p* < 0.01, respectively) compared to 0 h at the different time points following abiotic treatments.

**Figure 8 plants-12-01784-f008:**
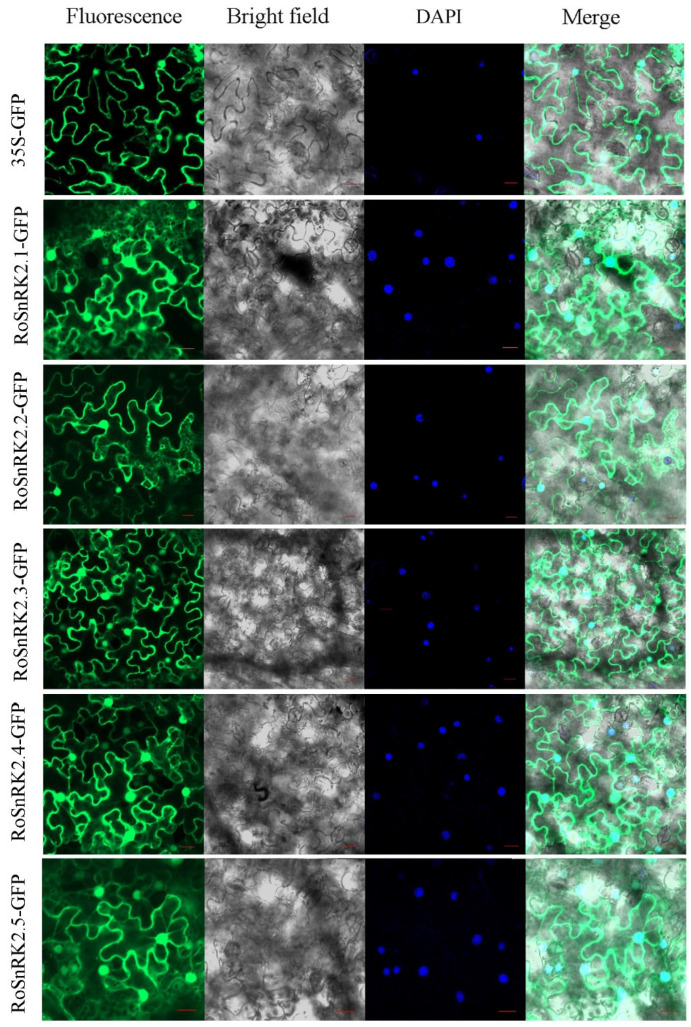
Subcellular localization of the fusion protein RoSnRKs-GFP in *N. benthamiana* leaves. The vector 35S-GFP was used as the control. Bar = 20 μm.

**Figure 9 plants-12-01784-f009:**
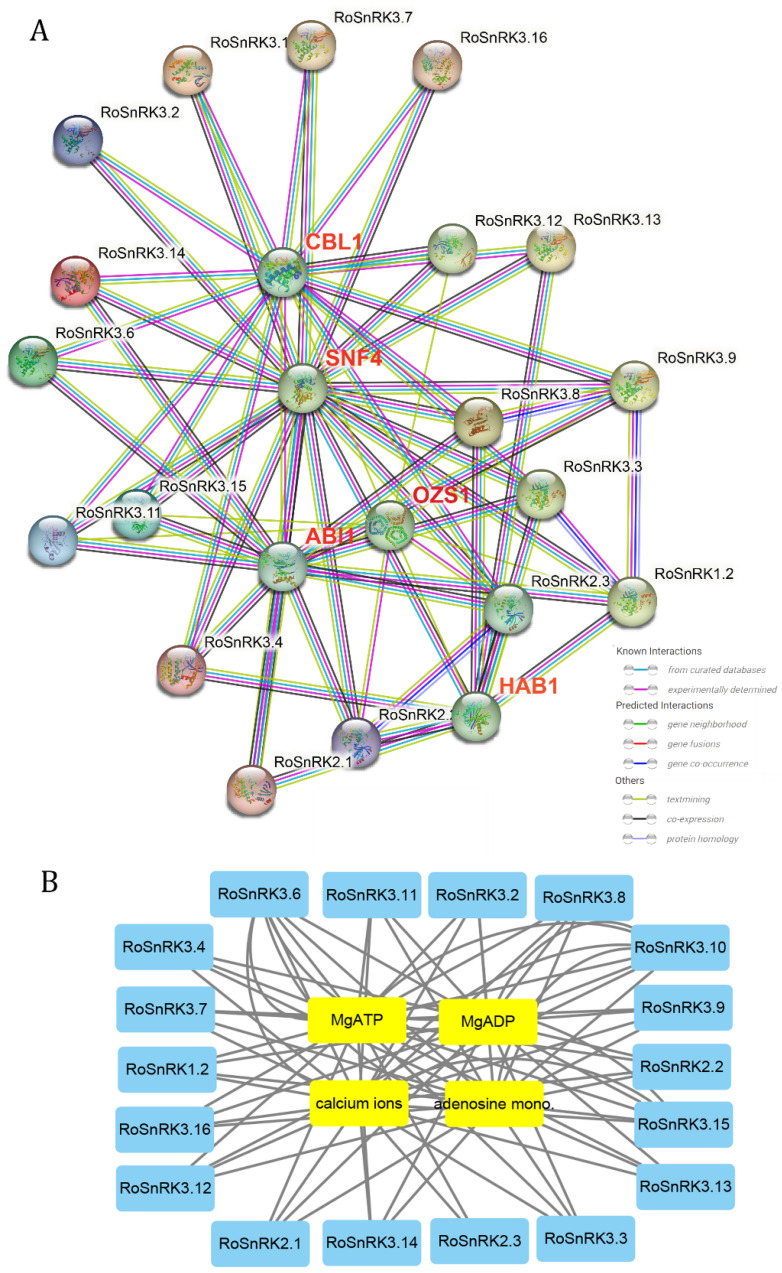
Interaction analysis of SnRK proteins based on predicted AtSnRK orthologs using the STRING and STITCH servers. (**A**) The protein–protein interactions, where RoSnRK proteins are marked with black font and predicted interacting proteins are marked with red font. (**B**) The protein–chemical interactions, where RoSnRK proteins are marked with blue boxes and predicted interacting chemicals are marked with yellow boxes. The networks were visualized using the Cytoscape software.

## Data Availability

The data presented in this study are available in the graphs and tables provided in the manuscript.

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
