# Peer review of "Comprehensive Genomic Analysis of SnRK in Rosaceae and Expression Analysis of RoSnRK2 in Response to Abiotic Stress in Rubus occidentalis"

_plants, 2023, doi:10.3390/plants12091784_

Round 1

Reviewer 1 Report

Authors identified 209 SnRK genes in seven Rosaceae species and did various in-silico analysis and expression analysis.

The Ms needs to be thoroughly revised before publication.

Ms seems to be very casually written. Most of the methods have not been described clearly, including identification of gene,  and expression section.

Title does not seems to be aligned with the Ms.

What is dispersed duplication?

Introduction needs to be revised with updated references like https://www.sciencedirect.com/book/9780323905947/plant-receptor-like-kinases

38 AtSnRK genes in Arabidopsis ? Sentence can not start with number. It should be ‘ A total of 38…

A figure for domain composition can be included in Ms.

Where the conserved motifs have been analysed? In gene or Proteins? Not clear from the results.  Authors should write gene structure and protein chareacterization in separate para.

Since the study is mostly in-silico, authors should also analyse protein-protein, Protein chemicals (www.mdpi.com/1422-0067/23/23/14867) and miRNA interaction can also be performed  following the suggested Ms (www.mdpi.com/2075-1729/12/7/941)

Most of the scientific names are not in format. Authors should thoroughly check.

I could not find any statistical parameters explained in the Ms even in RT PCR data.

How divergence time was calculated? Explain properly. Generally people use  the formula T = Ks/2r.

Discussion is very poorly written. There should not be any repeat from introction and results here. Authors should only discuss their results in conpariossion to other findings and what was their unique findings.

I could not see any concluding paragraph.

Author Response

  1. The Ms needs to be thoroughly revised before publication.  

Author’s response: I am very grateful to your comments on the manuscript. The manuscript has been modified again by Sophia Lee Guan, a native English speaker from University of Maryland, College Park. According with your advice, we have amended the relevant parts in the manuscript.

  1. Ms seems to be very casually written. Most of the methods have not been described clearly, including identification of gene, and expression section.

Author’s response: According to your kind suggestion, we have re-described the experimental method in detail and clearly. The description has been added on page 13 and 14 yellow highlights.

  1. Title does not seems to be aligned with the Ms.

Author’s response: According to your kind suggestion, we have modified the title to make it more reasonable “Comprehensive genomic analysis of SnRK in Rosaceae and expression analysis of RoSnRK2 in response to abiotic stress in Rubus occidentalis.”

  1. What is dispersed duplication?

Author’s response: In addition to whole-genome duplication (WGD), other modes of gene duplication are collectively deemed single-gene duplications (Panchy et al., 2016), including Tandem duplicates (TD), Proximal duplication (PD), Transposed duplication (TRD) and Dispersed duplication (DSD). Single genes can move, or be copied, from the original chromosomal position to a new position by various ways (Panchy et al., 2016; Qiao et al., 2019). Dispersed duplication (DSD) happens through unpredictable and random patterns by mechanisms that remain unclear, generating two gene copies that are neither neighboring nor colinear (Ganko et al., 2007; Qiao et al., 2019). The dispersed duplicates are prevalent in different plant genomes.

Ganko EW, Meyers BC, Vision TJ. Divergence in expression between duplicated genes in Arabidopsis. Mol Biol Evol. 2007;24:2298–309.

Panchy N, Lehti-Shiu M, Shiu S-H. Evolution of gene duplication in plants. Plant Physiol. 2016;171:2294–316.

Qiao X, Li Q, Yin H, Qi K, Li L, Wang R, Zhang S, Paterson AH. Gene duplication and evolution in recurring polyploidization–diploidization cycles in plants. Genome biology. 2019 Dec;20(1):1-23.

  1. Introduction needs to be revised with updated references like https://www.sciencedirect.com/book/9780323905947/plant-receptor-like-kinases

Author’s response: Thanks for your kind suggestion. According to your kind suggestion, we have removed old reference and updated the real-time references in introduction. to support our research topic. “The various reported protein kinase genes that are related to resistance include mitogen-activated protein kinase (MAPK) (Lin et al., 2021), calcium dependent protein kinase (CDPK) (Yip et al., 2019), receptor- like kinase (RLK) (Shumayla and Upadhyay, 2023; Karlik, 2023), and sucrose nonfermenting 1 (SNF1) - related protein kinase (SnRK) (Mishra et al., 2023)”. Please check it on page 1 yellow highlights and the section of reference on page 16.

Karlik E: Chapter 5 - Roles of plant receptor-like kinases in response to abiotic stress. In: Plant Receptor-Like Kinases. Edited by Upadhyay SK, Shumayla: Academic Press; 2023: 87-119.

Lin L, Wu J, Jiang M, Wang Y: Plant mitogen-activated protein kinase cascades in environmental stresses. International Journal of Molecular Sciences 2021, 22(4):1543.

Mishra S, Sharma R, Chaudhary R, Kumar U, Sharma P: SNF1-related protein kinase in plants: roles in stress response and signaling. In: Plant Receptor-Like Kinases. Elsevier; 2023: 195-209.

Shumayla, Upadhyay SK: Chapter 1 - An overview of receptor-like kinases in plants. In: Plant Receptor-Like Kinases. Edited by Upadhyay SK, Shumayla: Academic Press; 2023: 1-23.

Yip Delormel T, Boudsocq M: Properties and functions of calcium‐dependent protein kinases and their relatives in Arabidopsis thaliana. New Phytologist 2019, 224(2):585-604.

  1. 38 AtSnRK genes in Arabidopsis ? Sentence can not start with number. It should be ‘ A total of 38

Author’s response: Thank you for pointing out my non-standard writing style. According to your kind suggestion, we have revised these mistakes throughout the manuscript.

  1. A figure for domain composition can be included in Ms.

Author’s response: Thank you for your kind suggestion. According to your suggestion, the conserved domain of RoSnRK was analyzed by SMART. The corresponding results have been displayed in Fig. 5D, and the detailed description can be found on page 7 yellow highlights.

  1. Where the conserved motifs have been analysed? In gene or Proteins? Not clear from the results.  Authors should write gene structure and protein chareacterization.

Author’s response: According to your kind suggestion, we described the gene structures and protein characteristics in separate paragraphs, avoiding confusion. The analysis of conserved domains and motifs were described in the part of protein characteristics. Please check on page 6 yellow highlights for the corresponding description.

  1. Since the study is mostly in-silico, authors should also analyse protein-protein, Protein chemicals (www.mdpi.com/1422-0067/23/23/14867) and miRNA interaction can also be performed following the suggested Ms (www.mdpi.com/2075-1729/12/7/941)

Author’s response: Thank you very much for sharing your concern. According to your kind suggestion, we have analyzed protein-protein and protein-chemicals according to the methods described in the references (Kaur et al., 2022; Mendu et al., 2022). The corresponding results have been displayed in Fig. 9. The detailed description of methods and results can be found on page 15 and page 10-11 yellow highlights, and added the corresponding references on page 18.

Kaur A, Sharma A, Dixit S, et al. OSCA Genes in Bread Wheat: Molecular Characterization, Expression Profiling, and Interaction Analyses Indicated Their Diverse Roles during Development and Stress Response[J]. International Journal of Molecular Sciences, 2022, 23(23): 14867.

Mendu V, Singh K, Upadhyay S K. Insight into the roles of proline-rich extensin-like receptor protein kinases of bread wheat (Triticum aestivum L.)[J]. Life, 2022, 12(7): 941.

  1. Most of the scientific names are not in format. Authors should thoroughly check.

Author’s response: Thank you for pointing out my non-standard format. According to your kind suggestion, we have revised them throughout the manuscript.

  1. I could not find any statistical parameters explained in the Ms even in RT PCR data.

Author’s response: Thanks for your kind suggestion. We have added the describe of statistical parameters for qRT-PCR on page 15 “The results were calculated using Office 2010, and statistical analyses were performed by applying Student’s t-test at the levels of significant difference (p < 0.05 and p < 0.01 respectively).”, and Fig.7 figure legend on page 9 “Fig. 7 qRT-PCR analysis of RoSnRK2 genes under (A) NaCl and (B) ABA treatments. Standard errors and ANOVA were calculated by applying Student’s t-test. Single and double stars stand for the levels of significant difference (p < 0.05 and p < 0.01 respectively) compared to 0 h at the different time points following abiotic treatments”.

  1. How divergence time was calculated? Explain properly. Generally people use  the formula T = Ks/2r.

Author’s response: Thank you very much for sharing your concern. Generally, divergence time can be calculated according to the formula T = Ks/2r, (T = divergence time and r = divergence rate) was applied to compute the divergence time of duplicated genes. In the case of cereals, the value of r was proposed as 6.5 *10-9 ( Gaut, et al., 1996; Kaur et al., 2022) .

Gaut, B.S.; Morton, B.R.; McCaig, B.C.; Clegg, M.T. Substitution rate comparisons between grasses and palms: Synonymous rate differences at the nuclear gene Adh parallel rate differences at the plastid gene rbcL. Proc Natl Acad Sci. USA 1996, 93, 10274–10279.

Kaur A, Sharma A, Dixit S, et al. OSCA Genes in Bread Wheat: Molecular Characterization, Expression Profiling, and Interaction Analyses Indicated Their Diverse Roles during Development and Stress Response[J]. International Journal of Molecular Sciences, 2022, 23(23): 14867

  1. Discussion is very poorly written. There should not be any repeat from instruction and results here. Authors should only discuss their results in comparison to other findings and what was their unique finding.

Author’s response: Thank you for pointing out my poor writing style. Sorry for not describing it better written. As you suggested, we removed some of repeat viewpoints from introduction and results. In addition, we have re-written this section and provide a clear outcome in comparison to other findings. Please check the section of discussion on page 12

  1. I could not see any concluding paragraph.

Author’s response: According to your kind suggestion, we have added the section of “Conclusion” in concluding paragraph. The detailed description has been added in “Conclusion” on page 15 yellow highlights.

Thank you very much again for your kind suggestions. Thanks for your attention. Your comments significantly improved the quality of this manuscript.

Reviewer 2 Report

In the manuscript "Comprehensive genomic analysis of SnRK in Rosaceae and abiotic stresses responses in black raspberry (Rubus occidentalis L.)", the authors reported that a total of 209 SnRK genes were identified in seven Rosaceae species, allowing for inference of their phylogenic relationships, expansion, and evolution. The expression patterns of five RoSnRK2 genes in black raspberry were explored under salt and ABA treatments. Overall, this study provides a basis for elucidating the roles of RoSnRK2 genes in control of plant growth and response to abiotic stresses. It is an interesting and clear study with a valid and better selection of samples. The study included well-presented data and analysis, and tables and figures are clarified. However, minor revisions are needed as follows:

- Line 3 (title): Please change "stresses" to "stress".

- Line 3: I suggest using only the scientific name in the title.

- Lines 23, 27, and others: Please unify it as "abiotic stress", except for "biotic and abiotic stresses".

- Lines 41, 42, and others: Please unify "SnRK" in italic form, and any other gene terms, throughout the manuscript.

- Lines 51, 52: Please identify what is metabolic stress.

- Line 53: Please change "has" to "have".

- Line 63: "abiotic stress".

- Lines 76, 77, 78, and 99: "abiotic stress".

- Line 265: You can change "SnRKs" to "they" instead of looping.

- Line 393: Please include the relative humidity.

- Lines 394 and 395: The authors reported that "All raspberry seedlings were exposed to various stresses", reflecting stresses from more than two. please clarify.

- My recommendation is to "accept after minor revisions".

Author Response

 Author’s response:

I am very grateful to your comments on the manuscript. According with your advice, we have amended the relevant parts in the manuscript. Your suggestions are answered below:

- Line 3 (title): Please change "stresses" to "stress".- Line 63: "abiotic stress".- Lines 76, 77, 78, and 99: "abiotic stress".

Author’s response: Thank you for pointing out my non-standard writing style. According to your kind suggestion, we have revised these mistakes throughout the manuscript.

- Line 3: I suggest using only the scientific name in the title.

Author’s response: According to your kind suggestion, we kept only the scientific name in the title.

- Lines 23, 27, and others: Please unify it as "abiotic stress", except for "biotic and abiotic stresses".

Author’s response: Thank you for pointing out my non-standard writing style. According to your kind suggestion, we have revised these mistakes throughout the manuscript.

- Lines 41, 42, and others: Please unify "SnRK" in italic form, and any other gene terms, throughout the manuscript.

Author’s response: Thanks for your suggestion. We adhere to the writing principle of gene italics and protein non-italics, and modified throughout the manuscript.

- Lines 51, 52: Please identify what is metabolic stress.

Author’s response: Metabolic stress is a physiological process that occurs during exercise in response to low energy that leads to metabolite accumulation [lactate, phosphate inorganic (Pi) and ions of hydrogen (H+)] in muscle cells.

Halford NG, Hey SJ: Snf1-related protein kinases (SnRKs) act within an intricate network that links metabolic and stress signalling in plants. Biochem J 2009, 419(2):247-259.

Ramon M, Dang TVT, Broeckx T, Hulsmans S, Crepin N, Sheen J, Rolland F: Default activation and nuclear translocation of the plant cellular energy sensor SnRK1 regulate metabolic stress responses and development. The Plant Cell 2019, 31(7):1614-1632.

- Line 53: Please change "has" to "have".

Author’s response: Thank you for pointing out my non-standard writing style. According to your kind suggestion, we have revised these mistakes throughout the manuscript.

- Line 265: You can change "SnRKs" to "they" instead of looping.

Author’s response: According to your kind suggestion, we have replaced "SnRKs" with "they" in the appropriate position instead of looping.

- Line 393: Please include the relative humidity.

Author’s response: According to your kind suggestion, we have added the description of relative humidity “75% relative humidity”.

- Lines 394 and 395: The authors reported that "All raspberry seedlings were exposed to various stresses", reflecting stresses from more than two. please clarify.

Author’s response: Thanks for your suggestion. That was a wrong description, and we have revised it. “The raspberry seedlings were exposed to salt and ABA stresses, respectively. The plants were cultured in 200 mM NaCl for salt treatment, and 100 μM ABA for ABA treatment.”

- My recommendation is to "accept after minor revisions"

Author’s response: Thank you very much again for your kind suggestions. Thanks for your attention. Your comments significantly improved the quality of this manuscript.

Round 2

Reviewer 1 Report

The manuscript seems to be revised thoroughly. But some of the recent references mentioned in response are not in the Ms like Mendu et al 2022. 

Authors may also refer the Book "Plant receptor-like kinases; Role in development and stress" recently published by Elsevier having all details about the receptor-like kinases. 

Moreover the Ms has been improved and can be accepted after minor revision.

Author Response

The manuscript seems to be revised thoroughly. But some of the recent references mentioned in response are not in the Ms like Mendu et al 2022.

Author’s response: Thank you very much for your reminding, I am sorry for forgetting to insert the reference in the manuscripts, we have cited the reference (Mendu et al., 2022) in the corresponding place. Please check it on page 15 yellow highlights and the section of reference on page 18 reference [38, 39].

[38] Kaur A, Sharma A, Dixit S, Singh K, Upadhyay SK: OSCA Genes in Bread Wheat: Molecular Characterization, Expression Profiling, and Interaction Analyses Indicated Their Diverse Roles during Development and Stress Response. International Journal of Molecular Sciences 2022, 23(23):14867.

[39] Mendu V, Singh K, Upadhyay S K. Insight into the roles of proline-rich extensin-like receptor protein kinases of bread wheat (Triticum aestivum L.). Life, 2022, 12(7): 941.

Authors may also refer the Book "Plant receptor-like kinases; Role in development and stress" recently published by Elsevier having all details about the receptor-like kinases.

Author’s response: According to your kind suggestion, we have added detailed description refer the book in introduction. Please check it on page 1 yellow highlights “As well as signaling networks and developmental processes, plant protein kinases regulate stress response. As a result of understanding plant responses to diverse stresses, genetic engineering can be used to improve plant tolerability to abiotic stresses, thus enabling the advent of new agriculture practices [5, 6]."

[5] Shumayla, Upadhyay SK: Chapter 1 - An overview of receptor-like kinases in plants. In: Plant Receptor-Like Kinases. Edited by Upadhyay SK, Shumayla: Academic Press; 2023: 1-23.

[6] Karlik E: Chapter 5 - Roles of plant receptor-like kinases in response to abiotic stress. In: Plant Receptor-Like Kinases. Edited by Upadhyay SK, Shumayla: Academic Press; 2023: 87-119.